# Charge Carriers Density, Temperature, and Electric Field Dependence of the Charge Carrier Mobility in Disordered Organic Semiconductors in Low Density Region

**Seyfan Kelil Shukri [1,2,\*] and Lemi Demeyu Deja [1]**

[1] Department of Physics, Addis Ababa University, Addis Ababa P.O. Box 1176, Ethiopia; lemi.demeyu@aau.edu.et

[2] Department of Physics, JigJiga University, JigJiga P.O. Box 1020, Ethiopia

\* Correspondence: seyfan.kelil@aau.edu.et

**Abstract:** We investigate the transport properties of charge carriers in disordered organic semiconductors using a model that relates a mobility with charge carriers (not with small polarons) hopping by thermal activation. Considering Miller and Abrahams expression for a hopping rate of a charge carrier between localized states of a Gaussian distributed energies, we employ Monte Carlo simulation methods, and calculate the average mobility of finite charge carriers focusing on a lower density region where the mobility was shown experimentally to be independent of the density. There are Monte Carlo simulation results for density dependence of mobility reported for hopping on regularly spaced states neglecting the role of spatial disorder, which does not fully mimic the hopping of charge carriers on randomly distributed states in disordered system as shown in recent publications. In this work we include the spatial disorder and distinguish the effects of electric field and density which are not separable in the experiment, and investigate the influence of density and electric field on mobility at different temperatures comparing with experimental results and that found in the absence of the spatial disorder. Moreover, we analyze the role of density and localization length on temperature and electric field dependence of mobility. Our results also give additional insight regarding the value of localization length that has been widely used as $0.1b$ where $b$ is a lattice sites spacing.

**Keywords:** charge transport; charge carrier dependent mobility; temperature dependent mobility; spatial disorder

## 1. Introduction

Organic polymers that show semiconducting properties because of the weak $\pi$-bonding along the polymer chain have received widespread attention in the scientific community due to their easy manufacturing and possible applications in modern electronic devices: Organic light-emitting diodes (OLEDs) [1], organic field effect transistors (OFETs) [2–4], and organic solar cells [5]. The efficiency of organic disordered semiconductors (ODSs) based electronic devices depends mainly on the transport properties of charge carriers in the semiconductors. The central transport parameter is the drift mobility $\mu$ of a charge carrier which is much smaller than that in the inorganic counterparts. Consequently, for the best utilization of ODSs as an active material in electronic devices, the charge carriers transport properties have been widely studied [6–12], particularly, the dependence of mobility on temperature ($T$), electric field intensity ($F$), and charge carriers density ($p$) has been extensively studied both experimentally [13–20] and theoretically [6,9,21–27]. Yet, there is no consensus on an expression which can uniformly describe the characteristics of the charge carrier mobility in different disordered organic semiconducting polymers.

In ODSs the charge carriers are hopping between localized states that are thought to represent the conjugated polymer chain segments that are assumed to be distributed

randomly in space [6–10,16–18,22,23,25,26,28–30]. It is known that the energy of a charge carrier at different states is not the same, and assumed to have a Gaussian distribution [6]:

$$\rho(\varepsilon) = \frac{N}{\sqrt{2\pi\sigma^2}}\exp\left(-\frac{\varepsilon^2}{2\sigma^2}\right), \tag{1}$$

where $N$ is the total number of states, $\varepsilon$ is the energy of the charge carrier on a site relative to the center of the density of states (DOS) assumed to be zero in this case, and $\sigma$ is the energy distribution width that determines the amount of energy disorder whose value for ODSs is in the range from 0.05 to 0.14 eV [6,19,31,32]. The cause for the energetic disorder is believed to be the fluctuation in lattice polarization energies and the distribution of segment length in the $\pi$ or $\sigma$ bonded main chain polymers. The Gaussian shape of the DOS was assumed based on the Gaussian profile of the excitonic absorption band and by recognizing that the polarization energy is determined by a large number of internal coordinates, each varying randomly by small amounts [6]. The approximation made above on the sites energies distribution was verified experimentally using the direct measurement of the energies distribution in an electro-chemically gated polymer transistor [32].

The transport in ODSs occurs by a sequence of each charge carrier hopping from a localized occupied state $i$ to an unoccupied state $j$ with the transition rate expressed by Miller and Abrahams formalism [33]:

$$\nu_{ij} = \begin{cases} \nu_0 \exp\left(-\frac{2\mathbf{r}_{ij}}{\alpha} - \frac{\varepsilon_j - \varepsilon_i}{k_B T}\right) & \text{if } \varepsilon_j > \varepsilon_i, \\ \nu_0 \exp\left(-\frac{2\mathbf{r}_{ij}}{\alpha}\right) & \text{if } \varepsilon_j \le \varepsilon_i, \end{cases} \tag{2}$$

where $\mathbf{r}_{ij}$ is the displacement between the positions of a charge carrier before and after hopping. The coefficient $\nu_0$ is an intrinsic rate, which can be regarded as an attempt frequency that is determined by the tunneling (hopping) mechanism, and is simply assumed to be of the order of phonon frequency $\sim 10^{13}\text{s}^{-1}$, $\alpha$ is the localization length of the wave function of a charge carrier which is assumed to be the same for sites $i$ and $j$, $k_B$ is Boltzmann constant, $T$ is the temperature, and $\varepsilon_i$ and $\varepsilon_j$ are the energies of a charge carrier when at sites $i$ and $j$, respectively. The term $\exp\left(\frac{-2\mathbf{r}_{ij}}{\alpha}\right)$ is the overlap between the sites wave functions which decreases exponentially with the inter site distance $r$. The Boltzmann factor, $\exp\left(-\frac{\varepsilon_j - \varepsilon_i}{k_B T}\right)$, shows an activated process or the probability of the existence of a phonon of energy equal to $\varepsilon_j - \varepsilon_i$ which is either absorbed or emitted in order to conserve energy in the hopping process. There is no other activation energy except the difference in charge carrier energies between different sites that a carrier has to overcome in order to hop. The probability for hopping downward in energy is just $\nu_0 \exp\left(\frac{-2\mathbf{r}_{ij}}{\alpha}\right)$ by the principle of detailed balance with the premise that there are phonons that are always existing to absorb the energy difference between the final and initial states.

Bässler and coworkers [6] established an equation that expresses the temperature and electric field dependence of a charge carrier mobility in ODSs based on Monte Carlo (MC) simulations considering a Gaussian type distribution for on sites energies. According to these studies, the temperature dependence of mobility has non-Arrhenius form $\mu \propto e^{\left(\frac{-2\hat{\sigma}}{3}\right)^2}$, and the electric field dependence has a Poole–Frenkel mobility variation [34] $\mu = \mu_0 e^{\sqrt{\frac{F}{F_0}}}$ where $\hat{\sigma} = \frac{\sigma}{k_B T}$, $\mu_0$, and $F_0$ are material and temperature dependent constant parameters that hold in a limited field range in variance with that demonstrated experimentally by Gill in 1972 [13]. Following the equation established by Bässler and coworkers there have been continuing efforts [22,27,35–41] to construct an equation so that it can describe $\ln \mu \propto F^{1/2}$ [34] behavior over a much broader field range. The modifications have been made mostly on the spatial correlations between the energies on different sites and the solutions were found using analytical as well as MC simulation approaches.

In the equation formulated by Bässler [6] and Novikov et al. [22] the effects of charge carriers density was not considered in the analysis made to understand the depen-

dence of mobility on temperature and electric field. However, experimental studies, later on, showed that the charge carrier mobility is dependent on the density in conjugated polymers [10,14,15,18]. In these experiments, the mobility was measured for the same amorphous conjugated polymer used as an active material in OFETs and OLEDs. The results found show that the hole mobility of the film used in polymer FETs (PFETs) was larger than that used in polymer LEDs (PLEDs) by 3 orders of magnitude. The possible reasons suggested for the huge difference in mobility values obtained from this experiment was the large difference in charge carrier densities in these devices. In addition to the difference in the mobility values in both devices, the combined plots of mobility versus density shown indicate that the hole mobility is constant for the density less than $10^{16}$ cm$^{-3}$ and increases with a power law for densities greater than $10^{16}$ cm$^{-3}$. In these experimental setups of space-charge limited PLEDs it is difficult to separate the effects of density and electric field on mobility since both charge carriers density and electric field are simultaneously increased with the increase of the voltage across the thickness of the polymer film. However, in the analyses of experimental data the variation of electric field with the voltage (for low voltages less than 3 V) and, consequently, the mobility dependence on the field was not considered. In line with this, the experimental data for mobility as a function of density, for the unified diode and field-effect measurements, was analyzed for different temperatures on the basis of the equation derived by Vissenberg and Matters [42]. In this equation, only the density dependent mobility is included, the effect of field on mobility is disregarded, and the equation fails to reproduce the experimental current versus voltage characteristics at low temperatures and high voltages.

Cognizant of what is lacking in this analysis, Pasveer and coworkers [16] formulated an expression for the description of charge carrier mobility that encloses the effects of temperature, charge carriers density, and electric field based on the numerical solution of the master equation representing charge carriers hopping in a regularly spaced lattice sites using $\alpha = 0.1b$ (where $b$ is a lattice sites spacing) for the localization length value. The conclusion made is that the charge carriers density dependence of mobility is dominant at room temperature, while an electric field dependence of mobility is also important at low temperatures. In the approach of Pasveer et al. [16], the spatial disorder that arises from the variation of inter site wave functions overlap (off-diagonal disorder) was not incorporated. However, the applicability of this approach for hopping of charge carriers on randomly distributed states in disordered materials was challenged in recent literature [43,44] based on the comparison made between the results found for hopping transport on regularly spaced localized states and that for hopping transport on a randomly distributed localized state. The effects of spatial disorder of localized hopping states on the transport of charge carriers were presented in [27] using a kMC simulation approach considering uncorrelated Gaussian DOS. The spatial disorder was included using irregular Voronoi lattice that is formed by a shift of the states (sites) from the cubic lattice by a random value chosen from a Gaussian distribution with the spatial disorder parameter $\sigma_r$, and the analysis was made for $\sigma_r = 0, 0.05b$ and $0.1b$ (the value for localization length), where $\sigma_r = 0$ means for regularly spaced states. The results show that the spatial disorder changes the electric field dependence of mobility when compared with the results found for regularly distributed states, for different values of localization length, and also for different values of energy distribution width. The effect of charge carriers density on mobility was not covered. In other recent publications [19,20,24], the temperature dependence of mobility was shown to vary with density and localization length, respectively. The influence of temperature is more on the lower density, and on the lower localization length of charge carriers in ODSs of the same energy distribution width. So far, to the best of our knowledge, we have not seen MC simulation results reported to analyze the effects of density on mobility for hopping on randomly distributed sites that better represents the disordered materials than regularly spaced sites.

In this work we include the role of spatial disorder on charge carriers transport by considering randomly distributed hopping sites using a different approach from that used

in Refs. [6,27]. We design the model so that a state is randomly localized anywhere inside a sphere centered at regularly distributed sites. The radius of the sphere is related to the localization radius. This assumption is based on the premise that charge carriers, basically electrons, are localized in a certain region not strictly localized at one point. The size of the region is related to the width of the wave function of a charge carrier. The hopping is assumed to take place from a site placed in one sphere to a site in another sphere. For a sphere of radius $R$, the hopping distance from inside one sphere to another nearest neighbor sphere is in a range from $(b - 2R)$ to $(b + 2R)$ where $b$ is the distance between the centers of the neighboring spheres. With this model and assuming weak electron-phonon coupling, we used uncorrelated Gaussian distributed energies and the Miller–Abrahams rate equation to produce the required data using kMC simulations. We generate the data for mobility versus density at constant electric field ($\sim 5 \times 10^4$ V/cm) or less, for different density in the range from $10^{14}$ cm$^{-3}$ to $10^{17}$ cm$^{-3}$ at different temperatures less than or equal to room temperature, and analyze the effects of charge carriers density on mobility compared with the experimental results, and also with that found without incorporating the spatial disorder. The simulation results we got show that the mobility is also dependent on density in the lower density region. Our results for the lower density region is at variance with the density independent mobility results shown in [14,15]. In addition to this we present the effect of electric field on mobility in the low field region disregarded in [14,15,18] for different values of localization length. The role of localization length on the variation of mobility with other parameters are also presented and discussed. In general, we analyze our results in comparison with the previous results [14–16,20,27], and also propose that the value of localization length is close to $0.2b$ than to $0.1b$. The rest of the paper is organized as follows. In Section 2 we introduce the model and describe how the parameters that influence the transport properties are incorporated into the MC simulation methods. The main numerical results and discussions are presented in Section 3. Finally, a summary and conclusions are given in Section 4.

## 2. Methodology

The model system is a super cell of a hypothetical simple cubic lattice with a lattice parameter ($b$) in which each sub unit of a polymer chain is represented by a lattice point. These lattice points, known as states or sites, are localized on different conjugated polymer units (or segments). The distance from a site on one chain to any one of its neighboring sites on the same chain is different from that on another nearby lying chain(s), and the hopping process in a three-dimensional conjugated polymer system is anisotropic. As a consequence it is not easy to identify the microscopic parameters of a polymer film and use them precisely as the film is the aggregate of pattern-less and intermingled chains. Consequently, we also approximate the lattice parameter $b$, as it is done elsewhere [6,16,19,22,27,39], as an effective result of an average of inter chain and intra-chain lattice spacing. The geometry of this material can be considered as a three dimensional rectangular box of sides $L_x$, $L_y$, and $L_z$. An electric field is established by applying a voltage across the sample material. The charge carriers are injected to the sample polymer film, and the electric field forces the charge carriers to move so that their net direction of motion and electric field are the same if the charge carriers are holes and vice versa if electrons. In a real experiment, the density of charge carriers of ODSs is known to be varying in the range from $10^{14}$ cm$^{-3}$ to $10^{16}$ cm$^{-3}$ when used in PLEDs and from $10^{17}$ cm$^{-3}$ to $10^{19}$ cm$^{-3}$ in PFETs [14,15,18]. Taking this fact into consideration we introduce a certain number of charge carriers to the model polymer film in such a way that the charge carriers density variation looks like that described above for both PLEDs and PFETs. The direction of the electric field ($F$) is taken to be along the $x$-axis, which also becomes the direction along which the mobility is calculated. The magnitude of the electric field is approximated on the basis of the operating values of the potential difference between the electrodes in a specific arrangement of a device. The width of the conducting channel is in the $y$-direction, and the $z$-axis points in a direction perpendicular to $x$ and $y$-axes. Each position of the lattice array is assumed

to coincide with the center of a sphere of radius $R$. The sphere is assumed to represent the wave function of a charge carrier of localization length $\alpha \approx R$. In each sphere one hopping site is randomly placed. The distance between the centers of nearest neighboring spheres is the same as the lattice parameter $b$. However, the distance between the nearest neighboring hopping sites is a random variable between $b - 2R$ to $b + 2R$. In the case when the spatial disorder is ignored, the positions of sites coincide with the centers of spheres, and the minimum distance between hopping sites is $b = N^{-1/3}$, where $N$ is the concentration of localized states. The estimated value of the parameter $N$ for an organic conjugated polymer is between $N \approx 10^{20}\,\text{cm}^{-3}$ and $N \approx 10^{21}\,\text{cm}^{-3}$ [45]. To these sites $N_h$ number of charge carriers, so that the charge carriers density $\frac{N_h}{N}$ is in the required range, are introduced and randomly placed on different sites. The electric field along the $x$-axis due to the potential difference across the film ($V$) is assumed to be constant in the majority of the region where charge carriers move as verified recently in Ref. [46]. Therefore, the electric potential energy due to the electric field for a charge carrier at each site is:

$$U_{SD}(x,y,z) = Fe(0 - x),\tag{3}$$

where the minimum is fixed at the right edge of the sample polymer film, at $x = L_x$ in this case. Note that $x$ measures the distance along the field direction with a step distance of variable length that varies from $b - 2R$ to $b + 2R$. In the presence of an electric field, $U_{SD(i)}$ and $U_{SD(j)}$ are added to the energy $\varepsilon_i$ and $\varepsilon_j$ in Equation (2), respectively. In consistent with the GDM all sites to which a hop of a charge carrier is possible are assigned random energies sampled from a zero-centered Gaussian distribution function given in Equation (1) for a fixed value of distribution width ($\sigma$). Likewise in Refs. [6,16] these energies are assumed to be spatially uncorrelated.

The hopping rate from an occupied site $i$ to an empty site $j$, which are separated from each other by distance $r_{ij}$, is assumed to be governed by Miller and Abrahams formalism [33] given by Equation (2). Using this equation we determine the transition probability of a system of charge carriers from one configuration to another, and also the average time between successive transitions. We assume that double occupancy of the same site does not occur as it is not energetically favorable for the level of the charge carriers density we are now dealing with. We have also neglected the effects of the Coulomb interaction for the sample volume of a lower density less than or equal to $10^{17}\,\text{cm}^{-3}$. Here, the rectangular coordinates of all the sites are known in the code during the three dimensional sites formation. And the charge carriers are also identified with the names from 1 to $N_h$ given to them when they are first introduced to the sites. Therefore, the coordinates of each charge carrier are known at every moment in the code, and the magnitude of the distance between any two sites $\mathbf{r}_{ij}$ is:

$$r = \sqrt{(x_i - x_j)^2 + (y_i - y_j)^2 + (z_i - z_j)^2}.\tag{4}$$

In our simulation, we start the procedures by forming a three dimensional lattice of $L_x \times L_y \times L_z$ regularly-spaced positions which are the centers of spheres with the same radius $R$. In the second procedure, we assign an energy that a charge carrier can have at each site which is the sum of $\varepsilon(x,y,z)$ and $U_{SD}(x,y,z)$. In the third step we introduce $N_h$ number of charge carriers to the system by randomly placing them on the sites that are at randomly distributed positions in the spheres formed in the first procedure. $N_h$ correspond to densities between $10^{14}\,\text{cm}^{-3}$ to $10^{17}\,\text{cm}^{-3}$. In the fourth procedure, the exact three dimensional charge carriers distribution governed by the effective energy introduced above is realized. Initially, the charge carriers are distributed randomly throughout the total volume of the film. An equilibrium state of charge carriers distribution is assumed to be achieved in a certain number of Monte Carlo time steps (MCTSs). That is, we pick the first particle and check if its neighboring sites are empty or occupied. If there is no empty site we assign zero for its probability of hopping, and go to the second particle. However, if there is (are) neighboring unoccupied site(s), we calculate the change in energy

of the particle at a proposed neighboring empty site and initial site. Then we solve for the transition rate using Equation (2) excluding $\nu_0$ for each possible hopping site. We repeat this procedure for all charge carriers, and divide each transition rate to the sum of the total transition rates to find the probability of hopping for each possible path from the equation:

$$P_{ij} = \frac{\nu_{ij}}{\sum\limits_{i=1}^{N_h}\sum\limits_{j=1}^{l}\nu_{ij}}, \tag{5}$$

where $\nu_{ij}$ are the value obtained from Equation (2), $j$ is an integer that counts the possible nearest neighboring sites for each charge carriers, say from one to $l$, and $i$ counts the number of charge carriers from one to the last, say $N_h$. Equation (5) ensures that a detailed balance condition is satisfied in the steady state condition. Obviously, $\sum\limits_{i=1}^{N_h}\sum\limits_{j=1}^{l} P_{ij} = 1$ and, therefore, these probabilities generate a sequence of length intervals between 0 and 1. The sum of the probabilities for all possible paths is normalized to one, and the transition from one configuration to another takes place by one of the $N_h$ charge carriers along one path. This means that we use the probability length for each path and realize a transition of a system of charge carriers from one configuration to another by moving one of the charge carriers based on a stochastic approach. To decide for a particular hop we generate a random number $\omega$ from a uniform distribution in the interval between 0 and 1, which points at a particular interval and consequently at a particular destination site to which one of the the charge carriers jumps with the scheme [47]:

$$\sum_{i=0}^{N_h-1}\sum_{j=0}^{l-1} P_{ij} \leq \omega < \sum_{i=1}^{N_h}\sum_{j=1}^{l} P_{ij}, \tag{6}$$

where we have defined $P_{00} = 0$. We have restricted the possible destination sites $j$ in Equation (5) to the 26 nearest neighbor sites in an average volume of $3 \times 3 \times 3$ lattice sites. Even if there is a higher number of possible destinations involved, a vast majority of the hopping events occur to the nearest neighbor sites, and the effect of this restriction is negligible. This step is repeated for a certain number of MCTSs until we record enough data. One MCTS corresponds to moving one charge carrier after choosing from all charge carriers according to the procedure described above. Using an expression described in [6,48,49], the mean dwelling time $\tau$ of a charge carrier at site $i$ which is equivalent to the mean time for the transition that occurs between the states identified by the procedure described above is given as:

$$\tau = \frac{1}{\sum\limits_{i=1}^{N_h}\sum\limits_{j=1}^{26}\nu_{ij}}. \tag{7}$$

The transition time $\tau_h$ between states is distributed about the mean time $\tau$ is expressed as:

$$\tau_h = -\tau \ln(1-\xi), \tag{8}$$

where $\xi$ is a random number drawn from a uniform distribution between 0 and 1 different from that in Equation (6). This approach, which introduces a stochastic behavior in both the transition path and the time needed for the transition, ensures that the mean transition time is simply the mean of the inverse transition rates between these two states.

We let the system pass through $1 \times 10^5$ MCTSs to ensure that the system has reached a steady state distribution. It should be noted that different generations of the on-site energies from the GDM results in small but notable differences in the simulation results. The results are, therefore, averaged over 10 different generations of energy distributions. The fifth and final simulation procedure involves recording of the displacement of the

charge carriers along the length in the direction of electric field together with the time taken for this displacement to occur. The recording is performed for the last $1 \times 10^6$ MCTSs. The displacement of the charge carriers along the length in the direction of the electric field as a function of time gives the velocity, and the mobility of the charge carriers is obtained from the ratio of the velocity and electric field. This means that we obtain detailed information regarding the transport properties of charge carriers. Finally, simulation data are collected as a function of the charge carriers density, electric field, and temperature for different localization lengths.

## 3. Results and Discussion

Our calculation results are produced for charge carriers hopping conduction that takes place in a cubic sample of volume 200 nm × 200 nm × 200 nm with a periodic boundary condition. In the calculations we fixed the total number of sites, which are formed to have the corresponding concentration of $N = 10^{21}$ cm$^{-3}$, and varied the density ($p$) by varying the number of charge carriers. The mobility is expressed mostly in terms of lattice parameter $b$, disorder energy parameter $\sigma$, a unit charge on electron $e$, and phonon frequency $\nu_0$ (or in units of $\frac{b^2 \nu_0 e}{\sigma}$). When these parameters are known explicitly, the mobility is also expressed explicitly. The range of the magnitudes of electric field used for our calculations is chosen to highlight regimes of different transport behaviors. Most experimental data [13–16,41,50,51] fall, however, into the regime which is below low field strength (or less than or equal to $1 \times 10^5$ V cm$^{-1}$). The results we display below are obtained for different temperatures, two different values of localization lengths and corresponding spatial disorder parameter, and the on-site energy distribution width ($\sigma = 0.1$ eV) since the values for different disordered organic polymers as specified elsewhere [6,16,19,31,52] is in between 0.06 eV to 0.14 eV. The data generated for hopping on regularly distributed states ignoring the spatial disorder are also plotted together with that for randomly distributed states for two reasons; the first is to compare our results with the previously established results [16] for hopping on perfect lattice, and the second is to specify the difference between the MC simulation results found for hopping transport excluding the spatial disorder and that found with the inclusion of the spatial disorder. In all the figures where there are both broken and solid curves, the broken curves represent the mobility calculated for the hopping on randomly distributed states, and the solid ones represent the mobility calculated for the hopping on regularly distributed states excluding the spatial disorder. Our discussion will be based on the results that incorporate the spatial disorder except when we compare the results represented by broken and solid curves.

We first present a charge carrier mobility as a function of an electric field for different values of $\hat{\sigma} = \frac{\sigma}{k_B T}$ and also two different values of localization length ($\alpha$). The mobility is plotted on a logarithmic scale and the electric field is expressed in units of $\frac{\sigma}{eb}$ for $\hat{\sigma}$ in the range from 1 to 6, for two different values of localization length, $\alpha = 0.1b$ and $0.2b$, and for the same density as shown in Figure 1.

The curves in both panels of Figure 1 show that the mobility in the lower electric field region decreases with the increase of $\hat{\sigma}$ as verified and justified in the previous reports [6,14–16]. In the lower electric field region the mobility is more influenced by temperature than electric field, the transport mainly occurs due to diffusion and the mobility remains almost constant. At an electric field larger than $\frac{\sigma}{eb}$ the impact of activation barriers on a charge carrier hopping decreases with the increase of the electric field and the conduction is dominantly caused by the drift due to the field, and the mobility increases with the electric field until a saturation value is attained corresponding to each $\hat{\sigma}$ greater than 1. Whereas, for the lower value of localization length, $\alpha = 0.1b$, mobility increases with the electric field until it reaches saturation. The change becomes significant when $\hat{\sigma}$ gets larger. The saturated mobility values for different $\hat{\sigma}$ are nearly the same and reach at nearly the same electric field provided that $\alpha$ is the same. The reason for the occurrence of saturation is due to the fact that the electric field reduces the barrier height for an energetic uphill jumps from the transport energy level in the field direction and thereby

enhances the mobility in the hopping conduction of charge carriers as described in [34]. The drift mobility is calculated from the $\mu = v/F$ relation which leads to mobility reduction with the increase of electric field, where $v$ is the drift velocity of a charge carrier due to the electric field ($F$). This means that increasing the electric field is more effective in reducing the activation barrier, though it also reduces mobility, until the electric potential energy difference between any two sites separated by a distance $x$ along the electric field, $eFx$, approaches (but does not surpass) the energy scale ($\sigma$). If $eFx$ is greater than $\sigma$ an excess energy from $eFx$ above $\sigma$ which is the same as $eFx - \sigma$ does not have any contribution to speed up hopping conduction but reduces mobility since it appears in the mobility calculation, $\mu = v/F$. We can also interpret $eFx$ as an energy that reduces the activation barrier, and the increase of mobility with electric field is effective provided that the difference between $eFx$ and the average activation barrier height between hopping sites separated by a distance $x$ is significant. The reference energy value we took here, $\sigma$, to justify the statement that electric field raises mobility until $eFx$ reaches $\sigma$ may not be exact and needs experimental verification though it makes sense and seems reasonable. Due to this reason the effect of electric field in increasing the mobility is more significant for the larger $\sigma$. Similarly, temperature plays a crucial role in reducing the difference between the energies at the transport level and that of the hopping particle at the equilibrium level as discussed elsewhere [19,24]. Most of the solid (or broken) curves in Figure 1 show an overall similar behavior except that there is an increase of mobility with the increase of localization length. We also observe that the mobility saturates at a lower electric field in the case of a larger localization length. This indicates that a charge carrier conduction increases with the localization length and the effect of the electric field on mobility decreases with the increase of localization length. The possible justification for this is that when $\alpha$ increases the probability for a charge carrier hopping to a site at a larger distance increases. This gives rise to the increment of $eFx$, which will, consequently, increase the role of electric field to reduce the barrier height between hopping sites separated by a distance x. This can be the reason for the mobility saturation observed at relatively lower electric field than when the localization length is smaller. The charge carrier hopping rarely occurs when the localization length is small (or the state is strongly localized) at a relatively high $\hat{\sigma}$ in a lower electric field region. Another important point that we can infer from our simulation results is that the possible value of the localization length is closer to $0.2b$ than to $0.1b$. The argument here for this suggestion, though it needs experimental verification, is that the mobilities found are in the same order with that demonstrated experimentally [13–15] when the localization length is $0.2b$ than that found for $\alpha = 0.1b$ which is much lower.

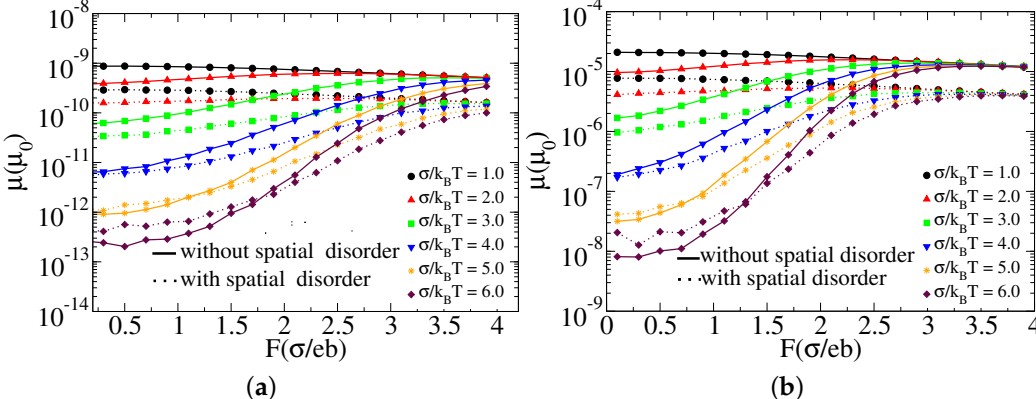

**Figure 1.** The logarithm of a charge carrier mobility $\ln(\mu)$ versus electric field ($F$) for different values of $\hat{\sigma}$ and $\alpha$ and for the same density ($p = 1 \times 10^{16} \, \text{cm}^{-3}$). (**a**) $\alpha = 0.1b$, (**b**) $\alpha = 0.2b$, where $b$ is a lattice parameter, and $\mu_0 = \frac{b^2 \nu_0 e}{\sigma}$.

Furthermore, when we compare each pair of broken and solid curves corresponding to each $\hat{\sigma}$ greater than a value of 2 we observe that the mobility variation with the electric

field is more pronounced for the solid curves than the broken ones. Besides this, we also see that mobility is larger for hopping on regularly distributed sites than that on randomly distributed sites except in the low field region when $\hat{\sigma}$ is less than 4. The curves for $\hat{\sigma} = 2, 3,$ and 4 in Figure 1, particularly in the first panel, show similar trends with that shown in [27] for randomly distributed sites with the same reduced disorder energy $\hat{\sigma}$ and localization length $\alpha = 0.1b$. In the low field portion for $\hat{\sigma} = 4, 5,$ and 6, each of the solid curves is below the corresponding broken curves, and we see that each solid curve crosses the corresponding broken curve at a certain value of electric field which testifies that the influence of electric field on mobility is more when hopping conduction is on regularly distributed sites than on randomly distributed sites. The reason for observing a solid curve above the broken curve for the same $\hat{\sigma}$ testifies that the spatial disorder adds additional hindrance on the hopping conduction of charge carriers for lower $\hat{\sigma}$. In the case when $\hat{\sigma}$ is large and the electric field is low, the favorable destination site for a charge carrier is a state with less energy at a closer distance. For hopping on randomly distributed states the distance between the states in the nearest neighbor spheres vary from $(b - 2R)$ to $(b + 2R)$, and there is a probability of finding a favorable destination state at a distance less than the mean value (or lattice parameter) which is zero for the hopping on regularly spaced states. This can be the reason for observing a solid curve below the broken curve for larger $\hat{\sigma}$ values in the low field region. In the case when $\hat{\sigma}$ is small, the activation barrier is small and a charge carrier can also hop to sites at larger distances. However, since there are sites at nearby distance the probability of hopping to sites in the nearby is large, and due to this a charge carrier which can hop to a site at larger distance chooses to jump to a site at a lesser distance when hopping on randomly distributed sites than that on regularly distributed sites. Similarly, the presence of sites at a distance less than the mean value reduces the influence of electric field on hopping of charge carriers on randomly distributed sites when compared to its influence on hopping of charge carriers on equally spaced sites of the same site spacing.

Our next simulation results displayed in Figure 2 together with that in Figure 1 give us additional insights on the role of electric field, energetic and spatial disorder, and localization length on a charge carrier mobility when the density of charge carriers changes from $1 \times 10^{16}$ cm$^{-3}$ to $1 \times 10^{15}$ cm$^{-3}$, keeping the rest parameters constant.

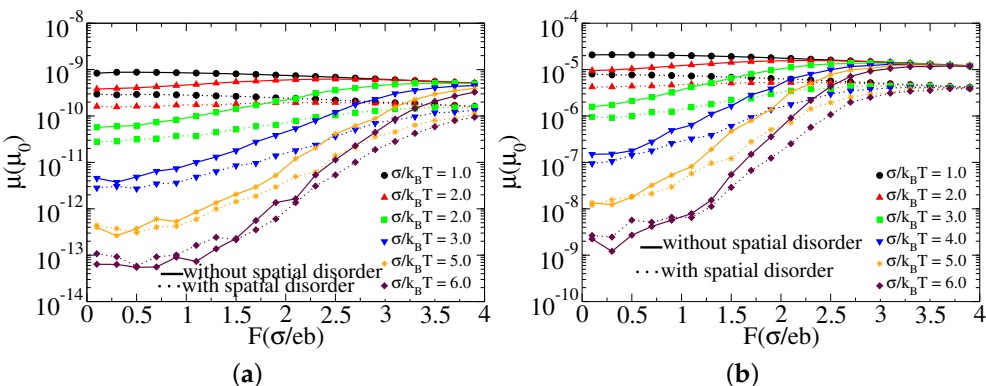

**Figure 2.** The logarithm of a charge carrier mobility $\ln(\mu)$ versus electric field $(F)$ for different values of $\hat{\sigma}$ and $\alpha$ and for the same charge carriers density $(p = 1 \times 10^{15}$ cm$^{-3})$. (**a**) $\alpha = 0.1b$, (**b**) $\alpha = 0.2b$, where $b$ is a lattice spacing, and $\mu_0 = \frac{b^2 \nu_0 e}{\sigma}$.

The curves in Figure 2 show the change of mobility with the electric field in a similar way that the corresponding curves in Figure 1 show regardless of the values of charge carriers density. Like that shown in Figure 1, the mobility increases with the electric field until the saturation value attained for $\hat{\sigma}$ is greater than 1. The increment becomes more pronounced when $\hat{\sigma}$ increases. We also observe an increment of mobility with charge carriers density though the density is low (in the range of the density for light emitting

diodes [14,15]). The reason for the increment of mobility with the charge carriers density is, as suggested elsewhere [27], due to the fact that the number of deep states, in the Gaussian density of states, that are filled with charge carriers increase with the increase of density. As a consequence more states with near lying energies become available for charge carriers hopping when the density gets higher. It means that at high charge carriers density the density of occupied states close to the Fermi energy gets larger, and more states within a small energy range are available as a hopping destination, and thus the activation energy barriers reduce their impact. In both Figures 1 and 2, we observe that the effect of the electric field on mobility gets stronger when $\hat{\sigma}$ gets larger except in the low electric field region. In the low field portion we see that mobility is independent of the electric field in Figures 1b and 2a when $\hat{\sigma} = 6$. The reason for not observing this behavior clearly in Figure 2b can be attributed to density and localization length difference. The electric field dependence of mobility decreases with density as suggested in [15]. The localization length in Figure 2b is larger than that in Figure 2a, and since the electrostatic energy $eFx$ is larger for larger localization length we see the changes of mobility with the field only for larger localization length. The results we obtained for larger values of $\hat{\sigma}$ are similar with that shown in [16] though there is a difference for $\hat{\sigma} = 6$ in the low density region which has similarity with that suggested in [15]. To understand the reason for the slight variation with the experimental ones, we have calculated mobility as a function of charge carriers density for different temperatures fixing the electric field; we will present and discuss more on this below.

The simulation results for a charge carrier mobility versus charge carriers density for fixed electric fields at $F = 0.1 \times \frac{\sigma}{eb}$ and $F = 0.05 \times \frac{\sigma}{eb}$ are presented in Figures 3 and 4, respectively. The mobility as well as the charge carriers densities are expressed (on a logarithmic scale) in units of $\mu_0 = \frac{b^2 v_0 e}{\sigma}$ and cm$^{-3}$ for two different values of localization length $\alpha = 0.1b, 0.2b$ and different temperatures.

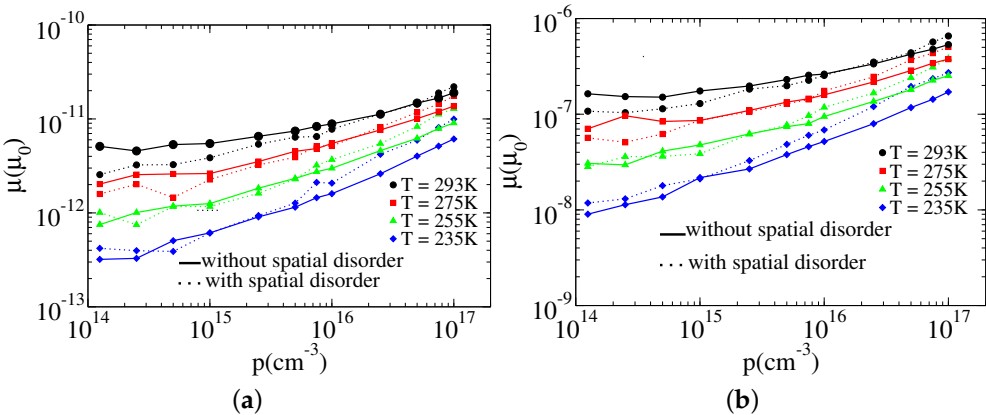

**Figure 3.** The logarithm of a charge carrier mobility $\ln(\mu)$ as a function of charge carriers density for the same disorder energy $\sigma = 0.1$ eV, different temperatures $(T)$, the same electric field $F = 0.1 \times \frac{\sigma}{eb}$. (**a**) $\alpha = 0.1b$, and (**b**) $\alpha = 0.2b$.

The curves results plotted in Figures 3 and 4 show that the increment of mobility increases with charge carriers density for both the lower and higher densities for all the temperatures specified here. However, at temperature $T = 235$ K the mobility is independent of density in the low density region in a narrow range between $10^{14}$ to $4 \times 10^{14}$ cm$^{-3}$. The reason for the increment of mobility with charge carriers density in a disordered Gaussian energy system is connected to the filling of the deep states due to, as described above, the impact that the density has on Fermi-energy, and also on the localization length. When the density increases the deep states are filled and the Fermi-energy are lifted up. Increasing charge carriers density can, also, increase the localization length [20]. These lead to an increment of the average hopping rate of a charge carrier,

which in turn leads to the increment of a charge carrier mobility. This means that mobility variation with the density presupposes the presence of deep states (or states with lower energies) and because of this the variation of mobility with density is more pronounced when, for the disorder energy, the energy distribution width ($\sigma$) is more or the temperature is less. Our simulation results found in the higher density region has similarity with the results demonstrated experimentally [14,15,18]. However, the results we got for the lower density region, though similar with the numerical results reported in [16], are at variance with the general conclusion made in [14,15] which says that mobility is independent of density in the low density region. The variation shown with the experimental results is due to the fact that the effects of electric field and density were not separated in the experiment. The effect of density on mobility, particularly, in the low density region can be compromised with that of the electric field which is visible when we examine the curves for two different electric fields shown in Figures 3 and 4. The charge carriers density dependence on mobility is more vivid in Figure 4 where the electric field is $F = 0.05 \times \frac{\sigma}{eb}$ than in Figure 3 where $F = 0.1 \times \frac{\sigma}{eb}$. The value of the electric field used in these experiments was close to the latter and because of this the influence of the charge carriers is suppressed.

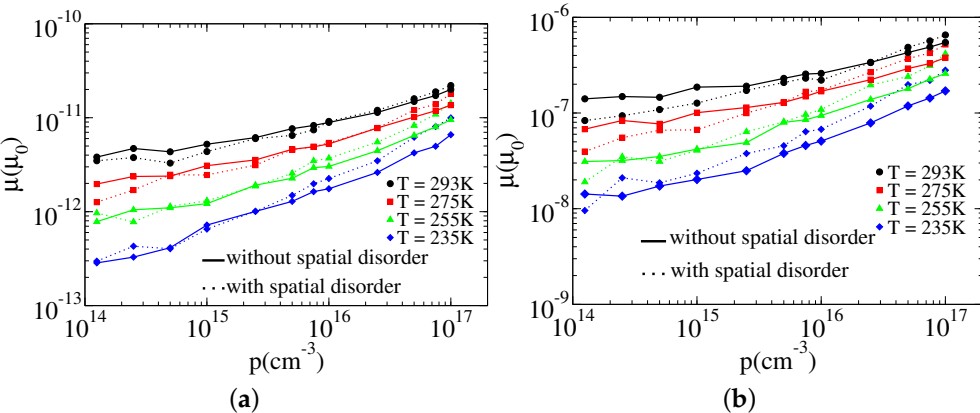

**Figure 4.** The logarithm of a charge carrier mobility $\ln(\mu)$ as a function of charge carrier density for the same $\sigma = 0.1$ eV, at different temperatures $T$ and $\alpha$ for the same electric field $F = 0.05 \times \frac{\sigma}{eb}$. (**a**) $\alpha = 0.1b$, and (**b**) $\alpha = 0.2b$.

When we compare the broken curves with the corresponding solid ones shown for each temperature in both Figures 3 and 4 we observe that the influence of density on mobility is stronger in the presence of the spatial disorder. Apart from this we also see the broken curves at the top of the corresponding solid curves at a low temperature and vice versa at high temperature in both figures for larger localization length, $\alpha = 0.2b$. And, in general, we see that each of the broken curves tends to move from the bottom of the corresponding solid curves to the top as the temperature changes from high to low, like that observed in Figures 1 and 2. A similar tendency is observed in the kMC simulation results presented in [27] for mobility (with and without spatial disorder) versus electric field at $\hat{\sigma} = 2$ when the localization length increases from $0.1b$ to $0.5b$ for spatial disorder $\sigma_r = 0.01b$, and fixed low charge carriers density. The possible justification for this was already presented above in connection with Figures 1 and 2.

We observed that charge carriers density reduces the influence of the electric field on mobility. Similarly, it determines also the dependence of mobility on an energetic disorder parameter or temperature. The graphs in Figure 5 support this claim for at least the lower density region. The graphs show similar behaviors for all charge carriers densities when $\hat{\sigma}$ is in the range between 1 to 3. However, for when $\hat{\sigma}$ is greater than 3 the mobility variation with $\hat{\sigma}$ depends on are different for different densities. The mobility decreases more with the increase of $\hat{\sigma}$ in the case of lower charge carriers densities than the higher ones qualitatively similar to that demonstrated in [19]. This reveals that in a hopping

transport the role of temperature is more when the charge carriers density is less, and also displays a transition from a non–Arrhenius to Arrhenius form of temperature dependence similar to that demonstrated experimentally [17,53].

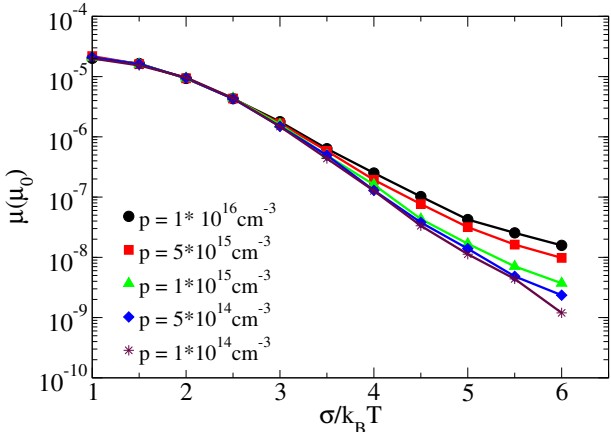

**Figure 5.** The logarithm of a charge carrier mobility versus disorder parameter $\hat{\sigma}$ in the range from 1 to 6 for different charge carriers densities $p = 1 \times 10^{14}\,\mathrm{cm}^{-3}$, $5 \times 10^{14}\,\mathrm{cm}^{-3}$, $1 \times 10^{15}\,\mathrm{cm}^{-3}$, $5 \times 10^{15}\,\mathrm{cm}^{-3}$, and $1 \times 10^{16}\,\mathrm{cm}^{-3}$ at the same electric field $F = 0.1 \times \frac{\sigma}{eb}$.

## 4. Summary and Conclusions

In this work, we have studied and presented a generalized view on the effects of specific materials and external parameters such as energetic and spatial disorders, charge carriers density, localization length, temperature, and applied electric field on the transport properties of charge carriers in disordered organic semiconductors using kMC simulation methods. Particularly, we have concentrated on the determination and analysis of identification of the effects of charge carriers density on mobility and also its influence on other intrinsic and extrinsic parameters dependence of charge carriers mobility. Our study was based on incoherent hopping of charge carriers between localized states that are regularly and randomly distributed in space. Using a Gaussian disordered model [6] and employing Miller and Abrahams formalism [33] for a hopping rate, we generated data and analyzed the dependence of mobility on temperature, applied electric field, and charge carriers density. We distinguished the effects of electric field from that of density and found results that show change of mobility with density also in the low density region. On the other hand, for hopping on regularly spaced sites in the low density region, the mobility change with density is found to decrease when with $\hat{\sigma}$ decreases and becomes constant for $\hat{\sigma}$ less than or equal to 3. At a higher $\hat{\sigma}$, for both regularly and randomly distributed sites, mobility increases with the increase of density. Consequently, we concluded that the effect of the disorder parameter ($\hat{\sigma}$) on mobility, in the low density region, is more pronounced than that of the density. Moreover, we observed that mobility is less for hopping on randomly distributed system than for that on regularly distributed system at low energetic disorder and vice versa for the larger energetic disorder. We also found that the dependence of density on mobility is stronger for hopping on randomly distributed states than on regularly distributed states. The generated data also yielded information on the localization length of the wave function of a charge carrier when analyzed in comparison with experimental data. For this parameter, different values have been used in the literature [6,16,54]. The value of the localization length ($\alpha = 0.1b$ used in [16]) seems small and could not lead to results mostly in the range between $10^{-7}\,\mathrm{cm^2 V^{-1} s^{-1}}$ that were demonstrated experimentally [13–15]. The consistency of our numerical data with the previous MC simulation and experimental results support the assumption we made to model a disordered system for the simulation studies, notably the statistical nature of the irregularly distributed hopping sites. On the basis of this agreement we can conclude that the material parameters that describe the charge carriers properties of the system, that is,

the charge carrier localization length, the energetic disorder, and the lattice sites spacing have been given appropriate values.

**Author Contributions:** Conceptualization, S.K.S. and L.D.D.; methodology, S.K.S.; validation, S.K.S. and L.D.D.; data curation, S.K.S.; writing original draft preparation, S.K.S.; writing review and editing, L.D.D..; visualization, S.K.S. and L.D.D.; funding acquisition, L.D.D. All authors have read and agreed to the published version of the manuscript.

**Funding:** This research received no external funding.

**Data Availability Statement:** The data that support the findings of this study are available from the corresponding author upon reasonable request.

**Acknowledgments:** We would like to thank the International Program in Physical Sciences, Uppsala University, Uppsala, Sweden for the financial support to fulfill the necessary research facilities for this work. We are indebted to Mulugeta Bekele for their valuable discussions and comments to the manuscript.

**Conflicts of Interest:** The authors declare no conflict of interest.

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
