# Peer review of "Charge Carriers Density, Temperature, and Electric Field Dependence of the Charge Carrier Mobility in Disordered Organic Semiconductors in Low Density Region"

_condensedmatter, doi:10.3390/condmat6040038_

Round 1

Reviewer 1 Report

The manuscript “Charge carriers density, temperature and electric field dependence of the charge carrier mobility in disordered organic semiconductors in low-density region” by Seyfan Kelil and Lemi Demeyu investigates the transport properties of disordered semiconductors using Monte Carlo simulations. The authors report mobility at lower densities.  The subject is timely, and the results are interesting. However, before the manuscript is ready for publication, the authors should make an effort to place their work in a better perspective with the existing literature. In particular, it would help to discuss the following papers in the manuscript:

Phys. Chem. Chem. Phys., 2018,20, 8897-8908

Phys. Rev. B 81, 045202 (2010)

PHYSICAL REVIEW B 71, 045214 (2005) 

Carbon 183, 774-779 (2021)

The figures are hard to read. The authors should try to enlarge the fonts in figures to make them more presentable.

Author Response

Point 1: The manuscript “Charge carriers density, temperature and electric field dependence of the charge carrier mobility in disordered organic semiconductors in low-density region” by Seyfan Kelil and Lemi Demeyu investigates the transport properties of disordered semiconductors using Monte Carlo simulations. The authors report mobility at lower densities. The subject is timely, and the results are interesting. However, before the manuscript is ready for publication, the authors should make an effort to place their work in a better perspective with the existing literature. In particular, it would help to discuss the following papers in the manuscript: Phys. Chem. Chem. Phys., 2018,20, 8897-8908 Phys. Rev. B 81, 045202 (2010) PHYSICAL REVIEW B 71, 045214 (2005) Carbon 183, 774-779 (2021)

Response 1:

We appreciate the reviewer for the constructive comments after thoroughly looked into our work. We found the comments very helpful and did our best to review our work using also the points and results presented in the references mentioned by the reviewer. The change made is included in the manuscript with different colours (red color represents the phrases that should be deleted and the added ones are designated by blue color).

Point 2: The figures are hard to read. The authors should try to enlarge the fonts in figures to make them more presentable.

Response 2: The comment is well taken, and the fonts in figures are modified so that one can easily read them.

Reviewer 2 Report

In this manuscript, the authors discussed about charge mobility in disordered organic semiconductors and found the importance of disorder parameter in the low density region. It is good practical working of simulation. They could model and calculate the mobility on a lower density region that was not possible so far. To do this, they developed MC simulation, that is their originality. From the manuscript, it is realized that disorder parameters are important in low density region. That is new findings for this topic area.  So it is acceptable for publishing in Condensed Matter.

Author Response

Point: In this manuscript, the authors discussed about charge mobility in disordered organic semiconductors and found the importance of disorder parameter in the low density region. It is good practical working of simulation. They could model and calculate the mobility on a lower density region that was not possible so far. To do this, they developed MC simulation, that is their originality. From the manuscript, it is realized that disorder parameters are important in low density region. That is new findings for this topic area.  So it is acceptable for publishing in Condensed Matter.

Response: We thank the reviewer for reading the manuscript thoroughly, and proposed the work for publication in Condensed Matter journal.

Reviewer 3 Report

This work (theoretical) focuses an interesting aspect regarding electrical carrier transport in disordered semiconductors (particularly organic ones). This topic has been studied since long time ago, and some different models was considered as more adequate to explain the electrical carrier hopping in (organic) semiconductor materials. The two more important models (and effectively in some agreement with experimental results) are based in Gaussian disordered hopping and Poole-Frenkel (modified) hopping, booth with different assumptions but in final with more or less similar macroscopic behavior.

This manuscript tries to develop a model taking into account the electrical carrier density.  Unfortunately, all the considerations are not fulfilled compared with actual state of the art and, in some aspects, are not really new, with some conclusions (correct) quite obviously. In fact, the results from this simple theoretical model (consider a “quantic” box – not new idea) only reveals the very old knowledge regarding this atomic / molecular electrical carrier hopping. In my opinion, this work can be really improved (see some considerations bellow) but cannot be accepted for publication in Condensed Mater journal.

As important keys (if the authors will whish to re-submitted a new manuscript containing the current idea) I can point the following:

  1. a) A very deep state of the art needs to be done. The average age of the cited works (references) has more that 10 – 15 years and (naturally) don´t discuss more recent work in this field;
  2. b) Consider follow the international (and recognized symbols) for physical amounts (for instance e is the electrical permittivity and not energy) and try to give a full physical description of several amounts (for instance x is cited (in page 5, equation 8) as a “random number drawn from a uniform distribution between 0 and 1 different from that in Eq. 6”. Although some explanation in the next sentence, the real physical nature of such random number is not clear. And this is only one example;
  3. c) It is not clear if the considered model (and in the final there are some confusions between energy states that are populated) takes into account the effect of energy levels acting as traps for the electrical carriers or only the frontier orbitals in the polymer chain. This issue should be clarified and (in my opinion) introduced in the model with realistic energy states densities and levels, as the electrical carrier transport in both low and high density depends – as clearly expected – on the space charge behavior;
  4. d) It is not acceptable the size of figures 1, 2, 3 and 4 and, more important, the size of the legends;
  5. e) Some figures (in particular figure 1) are cited in the text in a non-evident position (as well some equations). The authors should pay attention in the text location where the figures and equations are firstly cited;
  6. f) A clear discussion should be made. The authors, in “Summary and Conclusions” have non-sense sentences, with a not clear physical based explanation;
  7. g) Finally, a clear comparison with the (actual, not old) experimental results as well modified physical models, needs to be done.

Author Response

We have attached  the response of the comment point by point in the pdf file

This manuscript is a resubmission of an earlier submission. The following is a list of the peer review reports and author responses from that submission.

Round 1

Reviewer 1 Report

This manuscript presents results of Monte Carlo computer simulations of hopping transport in the framework of the traditional Gaussian disorder model with Miller-Abrahams hopping rates. Presentation of the research field is misleading. There is no novelty. Obtained results can hardly cause interest in the scientific community. Therefore, I cannot recommend publishing.

  1. There is no novelty in this manuscript. The same simulations with a=0.1b were performed in Ref.19 and the same simulations at low charge densities with a=0.2b are to be found in Ref. 8 (and references therein).
  2. In Sec. “Summary and Conclusions”, it is stated “Our study  was  based  on  the  fact  that the  charge  transport  is  due  to  incoherent  hopping  of  the charge carriers between localized states that are randomly distributed  in  ” However, not the “localized states that are randomly distributed in space”, but rather localized sates distributed on a perfect cubic lattice were studied in this paper.  

It has been proven in the literature that results for hopping transport on lattices have little to do with hopping transport via localized states that are randomly distributed in space.  Therefore, the results presented in the submitted manuscript are not applicable to materials with spatial disorder, such as disordered organic semiconductors.

3. Presentation of the research field does not correspond to the present level of research related to charge transport in disordered organic  semiconductors.

Author Response

Point 1. There is no novelty in this manuscript. The same simulations with α = 0.1b were performed in Ref. [1] and the same simulations at low charge densities with α = 0.2b are to be found in Ref. [2] (and references therein).

Response 1:

Effects of charge carriers density on mobility in ODSs were investigated experimentally first in Ref. [3] and later in Ref. [4]. The results show that the mobility varies with charge carriers density when the density is greater than 1016cm-3, and the mobility is independent of charge carriers density when the density is less than 1016cm-3.  As the effects of electric field and charge carriers density on mobility is not easy to separate in the experiment, in this paper we showed that mobility is dependent on charge carriers density also in the low density region provided that the disorder is strong, or temperature and/or electric field are low for α = 0.1b and α = 0.2b.

The reviewer mentioned that the same simulations with α = 0.1b were performed in Ref. [1] and for low densities with α = 0.2b in Ref. [2].

Yes, the effects of density on mobility was studied numerically in Ref. [1] only with α = 0.1b but not studied in Ref. [2] published in 1991; to our knowledge the experimental results on density dependent mobility in ODSs were reported first in 2003. Because of this we disagree with the reason mentioned here by Reviewer 1 for rejection.

Point 2 In Sec. ’’Summary and Conclusions’’, it is stated ’’Our study was based on the fact that the charge transport is due to incoherent hopping of the charge carriers between localized states that are randomly distributed in’’, However, not the ’’localized states that are randomly distributed in space’’, but rather localized states distributed on a perfect cubic lattice were studied in this paper.

It has been proven that in the literature that results for hopping transport on lattices have little to do with hopping transport via localized states that are randomly distributed in space. Therefore, the results presented in the submitted manuscript are not applicable to materials with spatial disorder, such as disordered organic semiconductors.

Response 2:

It is assumed that localized states distributed on a regularly spaced cubic lattice together with disorder parameters, disorder energy in this case, mimic the localized states in disordered organic semiconductors. It is based on similar approach that the equations in Refs. [1, 5] which have been used to analyze experimental data are constructed.

References

[1] W. F. Pasveer, J. Cottaar, C. Tanase, R. Coehoorn, P. A. Bobbert, P.W. M. Blom, D. M. de Leeuw, and M. A. J. Michels.
Phys. Rev. Lett., 94:206601, 2005.
[2] P. M. Borsenberger, L. Pautmeier, R. Richert, and H. Bässler. J. Chem. Phys., 94:5447, 1991.
[3] C. Tanase, E. J. Meijer, P. W. M. Blom, and D. M. de Leeuw. Phys. Rev. Lett., 91:216601, 2003.
[4] C. Tanase, P. W. M. Blom, and D. M. de Leeuw. Phys. Rev. B, 70:193202, 2004.
[5] S. V. Novikov, D. H. Dunlap, V. M. Kenkre, P. E. Parris, and A. V. Vannikov. Phys. Rev. Lett., 81:4472, 1998.

Reviewer 2 Report

Please check my comments in the attached file

Author Response

Dear Reviewer I have attached all the response point by point for all comments

Reviewer 3 Report

Please see attached word document

Author Response

Point. In summary, this work addresses an important topic. The approach is clear and the results are convincing. However, I do not find that it contributes a new result to the literature, particularly as it bears extremely close resemblance to results of variable range hopping from the late 1970s which has been extensively studied since then, and the author do not discuss this. The authors set out to explore mobility as a function of charge density, temperature, electric field, and localization length. The discussion on localization length does not show depth of understanding of why this parameter appears in the Miller Abrahams expression and their approach does not add to our understanding. The results on mobility as a function of temperature and electric field appear to be consistent with variable range hopping to the extent that one can judge from their results as they are presented. They do not add to our understanding. The efforts to understand mobility as a function of charge density fails to realize that this is an important parameter because of how it changes the density of states at the Fermi energy which is included in prior models. I do not find that this work adds new results or deepens our understanding of this topic.

Response:
As the viewer excellently expressed, Mott derived an expression for the charge transport conductivity as a function of temperature for disordered semiconductors starting from Miller Abrahams formalism considering constant density of states at the Fermi energy for a small energy range. This does not mean that this expression is binding for all ODSs since the density of states for disordered organic semiconductors has been justified to be close to Gaussian [1, 2], and the temperature dependent of mobility has been found to be different from that derived by Mott. Moreover, the temperature dependence of mobility even for ODSs has been identified to be affected by the density of charge carriers - which is non-Arrhenius type for low density and Arrhenius type [3] for high density. Similarly, the charge carriers density dependence of mobility in ODSs has been addressed in 2003 in Ref. [4] after Mott equation was established. The problem explored in this paper was that investigated experimentally in Refs. [4, 5] and opened for numerical verification. We solved the problem numerically based on variable range hopping model and discussed, to our knowledge, the results we got comparing with the previous experimental and theoretical results [1, 4–7]. The reason for mobility increment with density is explained connecting it with filling of deep states which can be seen as reducing the average energy barrier for hopping of charge carriers; this explanation was overlooked or not accepted by the reviewer. Because, he/she mentioned that we did not describe how the density of charge carriers change the density of states at the Fermi energy.

References
[1] H. Bässler. Phys. Stat. Sol.(b), 175:15, 1993.

[2] I. N. Hulea, H. B. Brom, A. J. Houtepen, D. Vanmaekelbergh, D. Vanmaekelbergh, J.J. Kelly, and E. A. Meulenkamp. Phys. Rev. Lett., 93:166601, 2004.

[3] N. I. Craciun, J. Wildeman, and P. W. M. Blom. Phys. Rev. Lett., 100:056601, 2008.

[4] C. Tanase, E. J. Meijer, P. W. M. Blom, and D. M. de Leeuw. Phys. Rev. Lett., 91:216601, 2003.

[5] C. Tanase, P. W. M. Blom, and D. M. de Leeuw. Phys. Rev. B, 70:193202, 2004.

[6] W. D. Gill. J. Appl. Phys., 43:5033, 1972.

[7] W. F. Pasveer, J. Cottaar, C. Tanase, R. Coehoorn, P. A. Bobbert, P.W. M. Blom, D. M. de Leeuw, and M. A. J. Michels. Phys. Rev. Lett., 94:206601, 2005.

Reviewer 4 Report

The study by Kelil and Demeyu investigates charge transport in disordered organic semiconductors in dependence of charge density, electric field, disorder and temperature. The interplay of the mentioned parameters is analyzed, and it is shown that charge density affects (increases) charge mobility stronger when the disorder is strong, or temperature and/or electric field are low. The study can be interesting for the readers of the Journal; however, the manuscript should undergo major revision prior to further consideration.

The two major concerns that should be addressed are:

  • First, the novelty of the work should be clearly emphasized. Description of the earlier work on the impact of charge density on charge mobility in Introduction should be significantly expanded. Similarly, analysis of the results in comparison to the previous ones should be performed in Results and Discussion section. Note that the phenomenon of trap filling is well-known and, to the best of my knowledge, extensively analyzed for disordered semiconductors (see e.g. Ref. [Phys. Chem. Chem. Phys., 2017,19, 7760-7771] as an example of recent studies).
  • Second, the presentation of the results should be qualitatively improved. The results should be organized so that the reader could obtain a clear picture of the interplay of various parameters on charge mobility. Repeats should be avoided. Summary and Conclusions section should be shortened.

The other concerns include:

  • Decrease of mobility with the electric field for low disorder (shown in Figures 1,2) should be highlighted and explained.
  • Several sentences and phrases should be rewritten since they are hard to understand, e.g.:

“Bässler and coworkers [8] did an extensive Monte Carlo (MC) simulations using Eq. (2) and uncorrelated on sites energies spectrum that has a Gaussian shape, known as a Gaussian Disorder Model (GDM), and established an equation that expresses the temperature and electric field dependence of a charge carrier mobility in ODSs[8].”

“…kMC simulation since there is a transition of state after each MC time step (MCTS) the charge carrier is forced to move even when it is in a deep state”

“…understanding of charge carriers mobility…”

“Moreover, our data for the mobility versus electric field for various charge carriers densities and localization lengths support this conclusion apart from that the mobility in the lower electric field region decreases with the increase of σˆ whereas for the higher values of σˆ the mobility increases with the increase of electric field until a saturation value is attained corresponding to each σˆ which are qualitatively similar with that reported in different literatures [8, 16, 19].” Should de divided in two or more sentences.

  • One passage seems to be incorrect: “The reason for the occurrence of this effect only for the larger localization length can be justified since the electrostatic energy eFx is larger for larger localization length and changes significantly with the field. “
  • Citation is needed for the range of disorder in ODS: “…for ODSs is in the range from 0.05 to 0.14 eV…”.
  • In the sentence beginning with “According to these studies the temperature dependence…”, definition of σˆ, μ0 and F0 ­should be given prior to the phrase “which holds in a limited field range…”
  • Language mistakes should be corrected.

Author Response

Dear editor I have attached the all comments one by one on pdf

Round 2

Reviewer 1 Report

Besides other issues, my main objection against publishing is related to the invalid lattice model used for the numerical study in this manuscript.  While the addressed materials are announced as “disordered organic semiconductors”, i.e., materials with spatial disorder, the modelled system is a perfect cubic lattice without spatial disorder. I mentioned in my previous report that results for hopping transport on lattices have little to do with hopping transport via  randomly distributed localized states.  The proof for this my statement can be found, for instance, in Phys. Rev. B 96, 195208 (2017) “Field dependence of hopping mobility: Lattice models against spatial disorder”, or in the review paper Phys. Stat. Sol. A  215, 1700676 (2018) “Mott Lecture: Description of charge transport in disordered organic semiconductors”.

In their reply to this my comment, the authors argue that the lattice model used in their study has been also previously assumed in several papers between 1991 and 2005. I think, however, that previous erroneous publications cannot justify publishing further erroneous papers. The invalidity of lattice models were, indeed, not known in 2005.  However, this invalidity has been well established since 2017. The science is developing, and there is no reason to repeat erroneous studies.

Reviewer 2 Report

In the following sentence there is for sure a mistake, I guess it has been written mobilty instead of density, please check it.

“that the mobility is dependent on mobility also in the lower density region which is at variance with the density independent mobility results shown in Refs.[17, 18] for lower density region.”

Reviewer 3 Report

Below please find the response from the author with my comments in bold

As the viewer excellently expressed, Mott derived an expression for the charge transport conductivity as a function of temperature for disordered semiconductors starting from Miller Abrahams formalism considering constant density of states at the Fermi energy for a small energy range.

This does not mean that this expression is binding for all ODSs since the density of states for disordered organic semiconductors has been justified to be close to Gaussian [1, 2],

In systems described by Mott Variable Range Hopping, the density of states (DOS) also has a Gaussian distribution as a function of energy just as found here in ODSs.  It seems that the authors are misinterpreting the constant density of states (DOS) in Mott Variable Range Hopping to mean constant as a function of energy, but it is actually indicating that the DOS is constant as a function of time

The reason it is important to specify that DOS is constant in time is to contrast it with the change in DOS in time that one finds in Marcus theory resulting from the reorganization energy.  When charge hopping events occur, the charge can cause the surrounding molecules to distort which causes a lattice polarization and lowers the site energy.   The reorganization energy describes how the energy changes with time when charge hopping events occur. 

In other words, the constant DOS in the Miller Abrahams/ Mott Variable Range Hopping expression is a constant in time, and typically has a Gaussian variation with energy just as in the ODS system described here.  

and the temperature dependent of mobility has been found to be different from that derived by Mott.

Mott Variable Range Hopping derives an expression for the variation in the electrical conductance with temperature, not the variation of the mobility with temperature.  The authors’ claim that the temperature dependence of the mobility derived by Mott is different from what is found here is not correct because Mott does not derive an expression for the mobility. 

It seems that the authors are extrapolating the variation of mobility with temperature for Mott from Mott’s variation of conductance with temperature.  This extrapolation is not valid, which is why Mott does not report the variation in mobility with temperature.  The mobility can be extrapolated from the conductance only when the system is Ohmic and satisfies the expression J= s E.  ODS and other similar disordered systems are by definition non-Ohmic because the application of an electric field changes the electronic potential landscape which then changes the hopping rate and thus changes the effective electrical resistance. 

Calculating mobility in disordered systems like ODS requires careful consideration of the fact that the system is non-Ohmic. 

Moreover, the temperature dependence of mobility even for ODSs has been identified to be affected by the density of charge carriers - which is non-Arrhenius type for low density and Arrhenius type [3] for high density. Similarly, the charge carriers density dependence of mobility in ODSs has been addressed in 2003 in Ref. [4] after Mott equation was established.

I find in Ref 4 that mobility as a function of electric field is reported, which is consistent with my earlier comments that mobility will change as a function of electric field in disordered systems and requires careful consideration.

In Ref 3, the field effect mobility as a function of temperature is reported which averts the problems that I describe above, and which I suggested in my first round of comments.

The fact that the mobility as a function of temperatures in ODSs switches between Arrhenius and non-Arrhenius is not inconsistent with a Mott VRH model.  As the charge density changes and traps are filled, the Fermi energy changes and you can find a transition to Arrhenius behavior.  Most systems described by VRH also show Arrhenius behavior.

The problem explored in this paper was that investigated experimentally in Refs. [4, 5] and opened for numerical verification. We solved the problem numerically based on variable range hopping model and discussed,

I did not find a discussion of the attempt to solve the problem based on variable range hopping.

to our knowledge, the results we got comparing with the previous experimental and theoretical results [1, 4–7]. The reason for mobility increment with density is explained connecting it with filling of deep states which can be seen as reducing the average energy barrier for hopping of charge carriers; this explanation was overlooked or not accepted by the reviewer.

I agree that the change in charge density will change the Fermi energy which in turn changes the electronic potential landscape i.e. the energy barrier for hopping. 

Because, he/she mentioned that we did not describe how the density of charge carriers change the density of states at the Fermi energy.

Reviewer 4 Report

I appreciate the consideration of most of my comments in the revised manuscript. I consider that it can be accepted for publication after accounting for several minor issues:

  1. Unfortunately, my suggestion for expanding the introduction with discussion of the earlier results was ignored. Although comparison of the results obtained with the previous studies was added in the Results and Discussion section, the suggested mentioning of these studies in Introduction appears to be desirable.
  2. There is a mistake in the added text in the phrase: "the mobility is dependent on mobility" (p.3)
  3. It remains unclear for me why "...in kMC simulation since there should
    be a hopping after each Monte Carlo time step (MCTS) the
    charge carrier is forced to hop even when it is in the deepest state". (p. 3) Why the charge carrier can not reside on the site? (the probability of this event equals the unity minus the total probability of hopping to various neighbors, isn't it?) This issue should be clarified, or the phrase should be deleted.